# Comparative Transcriptomics and Co-Expression Networks Reveal Tissue- and Genotype-Specific Responses of *qDTYs* to Reproductive-Stage Drought Stress in Rice (*Oryza sativa* L.)

**DOI:** 10.3390/genes11101124

**Published:** 2020-09-24

**Authors:** Jeshurun Asher Tarun, Ramil Mauleon, Juan David Arbelaez, Sheryl Catausan, Shalabh Dixit, Arvind Kumar, Patrick Brown, Ajay Kohli, Tobias Kretzschmar

**Affiliations:** 1Department of Crop Sciences, College of Agricultural, Consumer & Environmental Sciences, University of Illinois at Urbana-Champaign, Champaign, IL 61820, USA; tarun2@illinois.edu (J.A.T.); arbelaez@illinois.edu (J.D.A.); 2International Rice Research Institute, Los Baños, Laguna 4031, Philippines; s.catausan@irri.org (S.C.); s.dixit@irri.org (S.D.); a.kumar@irri.org (A.K.); A.Kohli@irri.org (A.K.); 3Southern Cross Plant Science, Southern Cross University, Military Rd, East Lismore NSW 2480, Australia; ramil.mauleon@scu.edu.au; 4Department of Plant Sciences, University of California Davis, One Shields Avenue, Davis, CA 95616, USA; pjbrown@ucdavis.edu

**Keywords:** co-expression network, drought-tolerant-yield, reproductive-stage drought, *qDTY*s, rice, transcriptomics

## Abstract

Rice (*Oryza sativa* L.) is more sensitive to drought stress than other cereals. To dissect molecular mechanisms underlying drought-tolerant yield in rice, we applied differential expression and co-expression network approaches to transcriptomes from flag-leaf and emerging panicle tissues of a drought-tolerant yield introgression line, DTY-IL, and the recurrent parent Swarna, under moderate reproductive-stage drought stress. Protein turnover and efficient reactive oxygen species scavenging were found to be the driving factors in both tissues. In the flag-leaf, the responses further included maintenance of photosynthesis and cell wall reorganization, while in the panicle biosynthesis of secondary metabolites was found to play additional roles. Hub genes of importance in differential drought responses included an expansin in the flag-leaf and two peroxidases in the panicle. Overlaying differential expression data with allelic variation in DTY-IL quantitative trait loci allowed for the prioritization of candidate genes. They included a differentially regulated auxin-responsive protein, with DTY-IL-specific amino acid changes in conserved domains, as well as a protein kinase with a DTY-IL-specific frameshift in the C-terminal region. The approach highlights how the integration of differential expression and allelic variation can aid in the discovery of mechanism and putative causal contribution underlying quantitative trait loci for drought-tolerant yield.

## 1. Introduction

Rice (*Oryza sativa* L.) is a staple crop feeding over half of the global population [1]. The Green Revolution accelerated the productivity of rice cultivation across Asia by focusing on irrigated, high input systems [2]. Intensification and expansion into new suboptimal cultivation areas, coupled with changing climatic conditions, however, necessitate a shift towards low input systems. Under resource- and water-limiting conditions, tolerance of both abiotic and biotic stress factors is crucial to ensure productivity.

Traditionally, rice was grown in areas naturally irrigated by seasonal floods [3]. Pre-domesticated rice is essentially a wetland species, making rice more sensitive to drought stress than most other staple crops [4]. Particularly, during the reproductive stage, drought typically causes yield reduction of 50% or more [5,6,7,8,9,10]. Consequently, water limitation is a major environmental constraint to rice production [11].

Successful strategies to identify factors contributing to drought tolerance involved mapping of quantitative trait loci (QTLs) for grain yield under drought conditions, so-called DTY (drought tolerant yield) QTLs [12]. By crossing the drought-tolerant donor N22 with Swarna, several major-effect DTY QTL, among them *qDTY1.1*, and *qDTY3.2* were identified as having consistent effects on grain yield under reproductive-stage drought stress (RDS) and no apparent yield or performance penalty under non-stress conditions [7]. These were subsequently introgressed into drought susceptible elite parents through backcrossing [13], resulting in the release of several drought-tolerant rice varieties. For example, “Bahuguni dhan-1”, a sister line of DTY-IL used in this study, was recently released in Nepal [13]. In addition, several other large-effect DTY QTLs were identified from other populations and utilized for their potential to confer drought tolerance [14,15,16,17]. Gene discovery work in *qDTY12.1* resulted in the identification of a NAM transcription factor as an intra-QTL hub gene [18,19].

To unravel specific drought responses in rice, transcriptome studies across different cultivars and drought stress conditions identified hundreds of differentially expressed genes (DEGs) in an organ- and time-specific fashion [1,10,20,21,22,23,24,25,26,27]. Collectively these studies pinpointed key transcription factors (TFs) involved in ABA-dependent and ABA-independent pathways to be upregulated during water deficit stress, effecting osmolyte production, reactive oxygen species (ROS) scavenging and ion transportation [10,22,28].

After an initial focus on studying drought during vegetative stages, the importance of RDS was soon realized [8,23,29,30,31,32,33]. From an applied perspective, grain yield under drought is the key trait, making yield-associated developmental processes and responses under RDS a focus of drought research in rice [8]. Knowledge of broad level biological mechanisms governing RDS responses, however, is still limited [34]. Convincing concepts on how drought-stress-related genes are regulated are still in their infancy. DEG analysis, based on single-genes, often failed to result in meaningful biological interpretations [35,36], prompting the development of network-based techniques that consider complex relationships among genes [37,38].

Gene co-expression networks (GCNs) are increasingly employed to explore system-level functionality of genes and have been found useful for describing the pairwise relationships among genes [39]. GCNs provide a structured pathway for extracting modular responses from large datasets that are often missed by DEG or ANOVA approaches [40]. It shifts the focus from single candidate genes to groups of related genes that likely operate together within a tissue or in response to a stimulus [41]. Genes clustering in a module, provide insight into potential regulatory functions [40,42,43]. An essential application of GCN analysis is to identify functional gene modules, which are a group of nodes that have high topological overlap [44]. Weighted gene correlation network analysis (WGCNA) can be used for co-expression network analysis of gene expression data to find modules of highly correlated genes [45]. In rice GCN analysis provided some insights into gene regulation under drought stress, including (i) consensus modules of downregulated and upregulated genes [46]; (ii) a module enriched for genes involved in water homeostasis and embryonic development, including a heat shock TF [47] and (iii) new candidates involved in drought response [48].

While a number of major QTL for DTY have been discovered, knowledge regarding the underlying physiological mechanisms is largely lacking. On the other hand, while a number of transcriptome studies provided some general insights into drought responses of rice, they did not take presence of specific DTY QTL into account. In the 2014 and 2015 drought field trials at the International Rice Research Institute (IRRI), a DTY introgression line (DTY-IL) performed well under drought without showing a penalty under irrigated conditions. We decided to investigate this line further in a comparative transcriptomic approach against its drought susceptible recurrent parent Swarna. Our rationale for this study was that combining a functional genomics with a classical genetics approach would improve resolution on drought tolerance mechanisms. By applying a gene co-expression network analysis and focusing on key source (flag-leaf) and sink (emerging panicle) tissues at reproductive stages, which had previously been demonstrated as critical for drought response [43], we speculated that adaptive mechanisms that drive yield under drought could be captured. By comparing the differential expression responses and overlaying genetic variation within the introgressed DTY QTL we further aimed to demonstrate that the differences at the genome-wide transcriptome level are modulated by the introgression segments. Our results support previous findings in respect to general mechanisms underlying drought response, and in addition suggest specific mechanisms underpinning DTY QTL.

## 2. Materials and Methods

### 2.1. Plant Materials and Experimental Treatments

Rice genotypes used in this study were: Swarna, a South Asian *indica* mega variety, susceptible to drought stress, and DTY-IL (designation IR 96321-1447-165-B-3-1-2), a highly drought-tolerant F_7_ introgression line containing N22 fragments in a Swarna background. DTY-IL is sister line of IR 96321-1447-651-B1-1-2, which was recently released as a drought-tolerant variety in Nepal [13].

Field experiments were conducted in the 2014 and 2015 dry season (DS), and were laid out in an augmented RCB design. The 2014 trial had 420 entries and 6 checks with 5 blocks in a single row per plot while the 2015 trial, had 46 entries and 2 checks with 4 blocks in four rows per plot. For both trials, only checks were repeated based on the number of blocks. For the drought screening, water was removed from the field around 28–30 days after transplanting by opening all drainage canals around the field. PVC pipes measuring 1.1 to 1.2-m long were installed in different parts of the field for water table measurements. The PVC pipes were placed 1 m below the soil surface. The water table was measured regularly starting 1 day after draining the field.

A pot experiment was arranged in a randomized complete block design with two treatments (well-watered and 2-weeks drought-stressed), two genotypes, and six replications in a greenhouse at the International Rice Research Institute (Los Baños, Philippines) from July to November 2015. Three pre-germinated seeds of the two genotypes were initially seeded on white porcelain pots filled with 15 kg of clean puddled field soil (not sterilized). Upon seedling establishment, a healthy seedling was retained in each pot and grown under a well-watered condition in the greenhouse until the booting stage before initiating a dry-down experiment. A day before imposing stress, all the pots were saturated with water and allowed to drain excess water for 24 h to maintain the field capacity (FC) so that the soil moisture amount in each pot was uniform. Then each pot was weighed to know the amount of water at FC. The temperature inside the greenhouse during the stress induction was at a maximum of 30–34 °C and a minimum of 23–26 °C, and a day-time relative humidity of 69%–95% (Appendix A). During the drought stress period, the pots were weighed daily, and the difference in weight on subsequent days was corrected by adding water to maintain the required FC [49]. For RDS, water was withheld at the reproductive R2 stage, on discrete morphological criteria as described by [50], until the soil moisture level dropped to 75% FC and was maintained for nine days, whereas control plants were well-watered. At day 10, FC was reduced to 50% for three more days (Appendix A). Flag-leaf and whole panicle samples of well-watered and drought-stressed treatments were collected at the R3 stage [50] on the 12th day of RDS and immediately flash-frozen in liquid nitrogen. Four independent biological replicates for each tissue and each genotype sample were harvested.

### 2.2. RNA Extraction and Sequencing

Total RNA was extracted using the Qiagen RNeasy Plant Mini Kit (Qiagen, Limburg, The Netherlands). RNA concentration was quantified using Nanodrop spectrophotometer (ND-1000; Nanodrop Technologies, Wilmington, DE, USA), while purity and integrity were established using an Agilent 2100 BioAnalyzer RNA 6000 Kit (Agilent Technologies, Santa Clara, CA, USA), with a RIN value of 8 used as quality threshold. Illumina library preparation and sequencing were completed following the standard protocols of Macrogen Inc. (Seoul, South Korea). Using Illumina HiSeq 2000 and HiSeq 4000 (Illumina, Inc., San Diego, CA, USA) platforms for the whole panicle and flag-leaf tissues, respectively, 101 bp aired-end sequencing was done. A quality check of raw RNA-Seq reads was performed using FastQC software (version 0.11.5) [51]. The sequencing data have been deposited in NCBI’s Gene Expression Omnibus (GEO) database under the accession number GSE145870.

### 2.3. Transcriptome Assembly and Expression Level Quantification

Raw fastq reads were filtered using Trimmomatic software, version 0.36 [52] using default settings. An indexed transcriptome fasta file was built from the rice japonica genome (IRGSP1.0) and annotation gff3 file from Rice Genome Annotation release 7 (RGAP 7, http://rice.plantbiology.msu.edu/), using the “gffread” function of Cufflinks (version 2.2.1) [53], and Salmon (version 0.7.2) [54] “index” function. Salmon then quantified transcript abundance in quasi-mapping mode directly using the indexed transcriptome and the trimmed high-quality paired-end reads with parameters “-l A, -seqBias and -gcBias” to allow automatic inference of library type, learn and to correct for sequence-specific and fragment-level GC content biases, respectively. Gene expression levels were normalized using the transcripts per million mapped reads (TPM) method, exported as estimated transcript abundance, and aggregated to gene-level expression using the Bioconductor package tximport (version 1.2.0) [55] complemented with the reader package (version 1.1.1), within the R environment (version 3.3.3). Principal component analysis (PCA) was performed in R using ggplot2 (version 2.2.1) and gplots package (version 3.0.1) to determine relationships between samples.

### 2.4. Differential Expression Analysis

*DESeq2* software (version 1.14.1) in R was used to identify DEGs in pairwise comparisons [56]. Two series of DE analysis using the following contrast arguments was performed with the reference assembly approach in flag-leaf and panicle samples: contrast 1—the condition effect for each genotype, in other words, IL_DvsIL_C, and SWA_DvsSWA_C, and contrast 2—the genotypic effect for each condition, in other words, IL_CvsSWA_C and IL_DvsSWA_D. Only genes that have at least ten reads in total were used for DE analysis. Differentially expressed genes (DEGs) were defined as those presenting an absolute fold change (FC) ≥ 2 or ≤ 0.5 and a False Discovery Rate (FDR) adjusted *p-*value ≤ 0.05 in any pairwise comparison. DEGs were then subjected to enrichment analysis of gene ontology (GO) terms, KEGG, and other metabolic pathways defined by a hypergeometric and Fisher’s exact test using agriGO (version 2.0), KOBAS (version 3.0), and STRING database (version 10.5). The MapMan tool (http://MapMan.gabipd.org) was used to visualize the involvement of the DEGs in pathways of interest.

### 2.5. Gene Co-Expression Network Analysis

Gene co-expression network analysis to group genes into modules used the R package WCGNA (v1.6.1). A power value of 6 and 9 for flag-leaf and panicle, respectively, predicted a gene co-expression network that exhibited scale-free topology with inherent modular features (Appendix A). The “blockwiseModules” function of WGCNA was used to detect and generate modules. Network interconnectedness was measured by calculating the topological overlap using the TOMdist function with a signed TOMType. Average hierarchical clustering using the “hclust” function was performed to group the genes based on the topological overlap dissimilarity measure (1-TOM) of their connection strengths. Network modules were identified using the dynamic tree cut algorithm (version 1.63.1) with minimum and maximum module size of 20 and 20,000, respectively, merging threshold function at 0.15, and deep split parameter set at level 2. The module “eigengenes” was used to calculate the correlation coefficient for the samples to identify biologically significant modules. To visualize the expression profiles of the modules, the eigengene (first PC) for each module was plotted using a customized bar plot function in R. To identify hub genes within the module, the module membership (MM) for each gene also known as module eigengene-based connectivity (kME) was calculated, based on the Pearson correlation between actual expression values and the module eigengene. Incorporating the gene significance (GS) measures, which could also be defined by the −log10 (*p*-value) from IL_DvsSwa_D contrast in DE analysis as external information into the co-expression network, genes within a module with the highest MM and GS values are highly connected within that module. To identify modules shared between the flag-leaf and panicle networks, a consensus network was generated. An in-house R function was used for overlap counting and statistical testing. The consensus network matrix for the flag-leaf and panicle tissue networks was plotted to show the significant overlap in gene count of two modules based on Fisher’s exact test with the −log10 (*p-*value).

### 2.6. Over-Representation Analysis (ORA)

The “kegga” function under the “goana” package in R utilized the user-supplied GO slim assignment and InterPro classification from RGAP 7 independently in the form of data.frame annotation alongside the multiple gene lists generated in each module. Gene.pathway was the data.frame linking genes to pathways and pathway.names was the data.frame giving full names of pathways. The universe was the vector specifying the set of unique gene identifiers used in WGCNA to be the background and not the whole genome. Finally, GO slim terms, and InterPro protein families and domains were called significantly over-represented in the gene set if the *p-*value is < 0.05.

### 2.7. Quantitative PCR Analysis

Primers were designed using QuantPrime (https://quantprime.mpimp-golm.mpg.de). cDNA was synthesized from 2 μg total RNA using the ImProm-II Reverse Transcription System (Promega, Madison, WI, USA), according to the manufacturer’s protocol. qRT-PCR was performed using two independent biological replicates and three technical replicates. qRT-PCR was set up in 386-well PCR plates with 0.2 μM primers using SYBR Green PCR Master Mix kit (Applied Biosystems, Foster City, CA, USA), following the manufacturer’s protocol in a reaction volume of 10 μL via a Roche LightCycler 480 Real-Time system (Rotkreuz, Switzerland). Reaction conditions were as follows: denaturation at 95 °C for 5 min, 45 cycles of 95 °C for 10 s, 60 °C for 15 s and 72 °C for 8 s, heating from 65 to 95 °C. Two internal reference genes ELF and ATU were designed to normalize the relative gene expression levels for flag-leaf and panicle tissue, respectively, using the 2^−△△^CT method with ΔCT = CTgene − CTreference gene [57]. For comparison of fold change, scatterplots were generated using the log_2_ fold change determined between RNA-Seq and qRT-PCR, which is defined as ΔΔCT (for comparative threshold cycle).

### 2.8. Genotyping

Using the C7AIR, 7098 SNPs were called for N22, Swarna and the DTY-IL [58]. Frequencies for each SNP across replicated samples were estimated and the most frequent genotype was considered true. Missing markers and monomorphic markers between the parents N22 and Swarna were discarded. For the remaining markers (1648 SNPs) genotypic calls from each parent were used to assign inheritance from N22 or Swarna. Fragments from the donor parent N22 were defined as consecutive SNPs with N22 homozygous genotypes. Markers that potentially represent miss-called double recombination were discarded if the probability of observing this event was smaller than 1 cM or 1 in 100 events. A graphical representation of the markers inherited from N22 and Swarna were graphed using the R package ggplot2. RNA-Seq reads of the candidate genes were aligned and visualized in IGV (version 2.8.2) relative to MSU7 (Nipponbare) and in the latest versions of the MH63 and N22. Clustal Omega (version 1.2.4) was used for multiple protein sequence alignments. FGENESH was used as the gene prediction tool with *O. sativa* vg. *indica* as the background organism. SNP-Seek (https://snp-seek.irri.org) was used to validate the NS SNPs across the 3 K genomes.

### 2.9. DNA Extraction and Sanger Sequencing

Genomic DNA was extracted from the leaves of N22, the DTY-IL *qDTY* donor, using the Qiagen DNeasy Kit (Qiagen, Limburg, The Netherlands) according to the manufacturer’s instructions. LOC_Os01g67030 (auxin responsive protein) including a 1.8 kb upstream promoter region was amplified using the forward primer GAGCGTGCAGTCCACTAGGCATTATC and reverse primer GTGACACGTATTCTGATGTACTG. The amplicon was cloned into the pGEM-T Easy Vector (Promega, WI, USA), as per manufacturer’s instructions and Sanger sequenced using Macrogen, Inc. South Korea as service provider.

## 3. Results

### 3.1. Phenotypes of Swarna and DTY-IL under Reproductive Stage Drought Stress

In the 2014 and 2015 dry season (DS) drought-stress field trials at IRRI, DTY-IL and Swarna, showed similar grain yield (Figure 1A) and plant height (Figure 1B) under irrigated conditions. Under reproductive-stage drought stress height was reduced in both genotypes to a similar degree (Figure 1B). While yield was reduced in both lines under drought, DTY-IL achieved about double the average grain yield compared to Swarna (Figure 1A). In addition, three weeks after draining the field, Swarna showed clear signs of leaf rolling. Leaf rolling was also observed for Swarna in the greenhouse experiment. While there was no visible difference in flag-leaf morphology between both genotypes under the well-watered condition (Figure 1C), a prominent leaf rolling phenotype in Swarna was observed after ten days of drought stress with complete leaf-rolling on the 12th day of drought stress (Figure 1D).

### 3.2. Generating a Transcriptional Map of the Moderate RDS Response in Rice

Mapping rates ranged from 77.7–92.9% for flag-leaf and 87.9–92.8% for panicle (Appendix A), covering 28,283 and 33,698 genes, respectively. Sample clustering and heatmap visualization of log_2_-transformed, normalized count data demonstrated clear separation between genotypes and treatments for both flag-leaf and panicle samples (Appendix A). Principal components (PC) analysis showed that the first and second PC explained 93% of the total variation for flag-leaf (Appendix A) and 81% for panicle tissue (Appendix A). Biological replicates of each genotype-treatment combination clustered together and the treatment effect was greater than the genotype effect for both tissues (Appendix A). An exception was a single Swarna panicle sample, which was removed from all further analyses as an outlier (Appendix A). The sum of the non-redundant/unique log_2_ normalized genes with significant changes of expression at FDR adjusted *p*-value ≤ 0.05 across four different contrasts were 17,616 and 18,614 for flag-leaf and panicle tissue, respectively. Pairwise DE analysis for all genotype-treatment combinations identified DEGs of significance for flag-leaf and panicle (Appendix A), and a Venn diagram was used to visualize three categories of unique and common responses in flag-leaf (Figure 2A and Appendix A, Table 1) and panicle (Figure 2B and Appendix A, Table 1).

#### 3.2.1. Expression Profiles of Drought-Responsive Genes in the Flag-Leaf Tissue under RDS

A total of 4180 genes were found to be drought-responsive in flag-leaves of both Swarna and DTY-IL (Figure 2A, Table 1). Gene ontology (GO) enrichment analysis showed functional enrichment among upregulated DEG for transcription, regulation of biological processes, and oxidation-reduction (Appendix A), while the shared downregulated DEGs were mainly associated with transmembrane transport, localization, and post-translational protein modification (Appendix A).

In Swarna flag-leaves a total of 515 (188 up- and 327 downregulated) genes were uniquely drought-responsive (Table 1). While no significant GO terms were detected for upregulated DEGs (Table 1), significant GO terms for uniquely downregulated DEGs were largely related to post-translational protein modification, photosynthesis, defense response, and programmed cell death (Appendix A). Pathway enrichment suggested photosynthesis, ubiquinone, and other terpenoid-quinone biosynthesis, as well as glutathione-mediated detoxification II and tyrosine biosynthesis (Appendix A), to be significantly downregulated. MapMan visualization supported the downregulation of cell wall, carbon metabolism, secondary metabolism, and light reaction in Swarna flag-leaves (Figure 3A and Appendix A).

In DTY-IL flag-leaves 108 (74 up- and 34 downregulated) DEGs were uniquely drought-responsive. While no significant GO terms could be associated with downregulated DEGs, oxidation-reduction, response to stress, and response to stimulus were among the significant GO terms in upregulated DEGs (Appendix A). Pathway enrichment suggested phenylpropanoid biosynthesis, dhurrin, xylan, and scopoletin biosynthesis, as well as detoxification of reactive carbonyls in chloroplasts to be uniquely upregulated under RDS (Appendix A). This was supported by MapMan visualization, showing upregulation of cell wall, lipids, and secondary metabolism (Figure 3B and Appendix A).

Numerous overrepresented *cis*-elements were found in the group of 327 promoters of uniquely downregulated DEGs in Swarna in flag-leaf, which are mostly involved as binding sites for dehydration responsive genes. These include ACGTATERD1, IBOX, PREATPRODH, MYCATERD1, MYCATRD22, CCAATBOX1, and MYB2AT (Appendix A-1). The overrepresented motifs in a set of 74 promoters in the uniquely upregulated DEGs in IL in flag-leaf were mostly functioning upon induction of dehydration stress through the ACGTATERD1 motif (Appendix A-2).

#### 3.2.2. Expression Profiles of Drought-Responsive Genes in the Panicle Tissue under RDS

A total of 4799 genes were found to be drought-responsive in panicles of both Swarna and DTY-IL (Figure 2B; Table 1). Enriched GO categories of the upregulated DEGs related to post-translational protein modification and response to stress (Appendix A), while enriched GO terms of the downregulated DEGs were mostly related to transmembrane transport, carbohydrate metabolic process, and localization (Appendix A).

In Swarna panicles, a total of 487 (184 up and 303 downregulated) genes were found uniquely drought-responsive (Table 1). No significant GO terms were identified within the uniquely upregulated DEGs. For the uniquely downregulated genes, significant GO terms included oxidation-reduction as well as monooxygenase activity, tetrapyrrole, and heme-binding (Appendix A). Significantly enriched pathways associated with downregulated DEGs were related to DNA replication, diterpenoid biosynthesis, phenylpropanoid biosynthesis, and photosynthesis (Appendix A). MapMan visualization supported the downregulation of cell wall, lipids, secondary metabolism, amino acids, as well as carbohydrate metabolism (Figure 3C and Appendix A).

In DTY-IL panicle 164 (108 up and 56 downregulated) DEGs were uniquely drought-responsive (Table 1). No significant GO enrichment was identified among the uniquely downregulated DEGs of DTY-IL. Prevalent GO terms of upregulated DEGs were related to protein amino acid phosphorylation as well as oxidation-reduction and carbohydrate metabolic processes (Appendix A). The most significantly enriched pathways in the panicle tissue of DTY-IL upregulated DEGs were related to propanoate metabolism, methylerythritol phosphate pathway, diterpenoid biosynthesis, camalexin biosynthesis, and circadian rhythm in plants (Appendix A). This was supported by MapMan visualization, showing upregulation of cell wall, lipids, secondary metabolism as well as amino acids (Figure 3D and Appendix A).

Overrepresented *cis*-acting elements in the group of 303 promoters of the uniquely downregulated DEGs in Swarna in panicle tissue mostly involved in dehydration response like the ACGTATERD1, MYBCOREATCYB1, ABRELATEDRD1, and IBOX motifs (Appendix A). Overrepresented *cis*-elements in a set of 108 promoters of uniquely upregulated DEGs in DTY-IL in panicle tissue were mostly related to dehydration response like the MYB2AT, MYBCOREATCYCB1, and ACGTABOX (Appendix A).

### 3.3. Drought-Stress-Related Gene Modules within the Transcriptional Map

WGCNA identified 21 distinct co-expressed modules with different expression patterns in flag-leaf (designated as FL-M1 to FL-M21, capturing 17,616 genes) (Appendix A), and 23 distinct modules for panicle network (designated as P-M1 to P-M23, capturing 18,614 genes) (Appendix A).

More than 70% of genes were distributed in the FL-M1 and FL-M2, and P-M1 and P-M2 (common response between DTY-IL and Swarna under RDS and control) for flag-leaf and panicle networks, respectively (Appendix A), signifying a common response shared between Swarna and DTY-IL. Modules FL-M1 and FL-M2 in the flag-leaf network showed “localization” and “transport” as the most enriched GO terms (Appendix A). In the panicle network, P-M1 genes were enriched for functions related to “RNA processing” while the P-M2 module was linked with “localization” and “transport” (Appendix A).

#### 3.3.1. Flag-Leaf Specific Modules

Two specific flag-leaf modules with a contrasting pattern of expression were investigated in more detail for implications in the differential performance of the two genotypes under RDS. FL-M14 consisting of 140 genes (Appendix A) had significantly higher expression profiles in all samples of the DTY-IL genotype under RDS, whereas a lower expression across the three other groups of samples was observed (Figure 4A). FL-M16 consisting of 102 genes (Appendix A) had significantly lower expression in Swarna under RDS, whereas the three other groups of samples had a higher expression (Figure 4B).

FL-M14 enriched BP GO terms were “cellular amino acid biosynthetic process” as well as “cell wall organization or biogenesis” (Appendix A). The enriched pathway included “biosynthesis of secondary metabolites”, “fatty acid elongation”, and “phagosome” while the overrepresented Interpro domains included “Expansin” as well as “Glycosyl transferase, family 43”, and “Plant peroxidase” (Figure 5A). FL-M14 hub genes included amidase and expansin (Appendix A; Appendix A). Interestingly, numerous cell wall organization and biogenesis genes are upregulated in DTY-IL and downregulated in Swarna under RDS (Figure 6).

FL-M16 was enriched in “oxidation-reduction process” as well as “cellular carbohydrate metabolic process” (Appendix A). The enriched pathways include “photosynthesis”, “folate biosynthesis”, and “vitamin B6 metabolism” while the overrepresented Interpro domains in FL-M16 included “Ferrodoxin—NAP reductase”, “Multicopper oxidase”, and “Photosystem antenna protein-like” (Figure 5B). FL-M16 hub genes included OsSub37—putative subtilisin homologue and cytochrome P450 (Appendix A; Appendix A).

#### 3.3.2. Panicle Specific Modules

Two contrasting panicle modules (P-M10 and P-M15) were investigated in more detail. P-M10 consisting of 138 genes (Appendix A) and P-M15 consisting of 73 genes (Appendix A) had significant interaction with drought response in a genotype-specific manner. P-M10 had significantly higher expression profiles in all samples of DTY-IL under RDS and a lower expression pattern across the other three groups of samples (Figure 7A). P-M15 had significantly lower expression profiles in all samples in Swarna under RDS, whereas the three other groups of samples had a higher expression (Figure 7B).

P-M10 enriched BP GO terms were “calcium ion transmembrane” as well as “lipid biosynthetic process” (Appendix A). Pathway enrichment analysis included “glutathione metabolism”, “phenylpropanoid biosynthesis”, and “ribosome” while the overrepresented Interpro domains were “Protein kinase” as well as “Plant peroxidase” (Figure 8A). P-M10 hub genes included OsWAK receptor-like cytoplasmic kinase and a serine-type peptidase (Appendix A; Appendix A). Analysis of genome-scale metabolic pathways in the DTY-IL revealed the up-regulation of genes involved in the biosynthesis of antioxidant enzymes and metabolites (Figure 9 and Appendix A).

P-M15 were GO enriched for “carboxylic acid metabolic process” as well as “coenzyme metabolic process” (Appendix A). The enriched pathways includes “carbon metabolism”, “glutathione metabolism”, and “pentose phospate pathway” while the overrepresented Interpro domains included “Cysteine synthase” and “Thiolase-like, subgroup” (Figure 8B). P-M15 hub genes included the glycosyl hydrolase family 29, and dehydrogenase/reductase (Appendix A; Appendix A).

The consensus network from the flag-leaf and panicle transcriptomes contained few significant overlaps in module classifications between the flag-leaf and panicle networks, consistent with the tissue-specific expression under RDS [23] (Appendix A). Similarly, colored modules between the flag-leaf and panicle networks contained a few significant overlaps of genes with a common consensus network module, consistent with their similar eigengenes profiles (Appendix A). The significant overlap portrays a common expression pattern for each condition of both genotypes (FL-M1 and P-M1, and FL-M2 and P-M2) and a common expression pattern for each genotype on both conditions (FL-M6 and FL-M7, and P-M5 and P-M6) in the flag-leaf and panicle networks, respectively.

### 3.4. Validation of Differential Gene Expression

RNA-Seq results were validated using ten genes from different response categories (increased, decreased, and non-differentially expressed genes upon treatment in both flag-leaf and panicle) for qRT-PCR (Appendix A). Three selected genes from within the *qDTY1.1* region (LOC_Os01g66120, LOC_Os01g66820, and LOC_Os01g67030) showed differential expression. LOC_Os01g66120 (no apical meristem protein) was upregulated in both genotypes and both tissues under drought. LOC_Os01g66820 (inactive receptor kinase At1g27190 precursor) was downregulated in DTY-IL flag-leaves under RDS. LOC_Os01g67030 (auxin responsive protein) was upregulated under RDS in DTY-IL panicles and downregulated in Swarna panicles but not affected by RDS in flag-leaves. A high correlation between qRT-PCR results and RNA-Seq results was observed for flag-leaf (*R*^2^ = 0.88) and panicle (*R*^2^ = 0.91) tissues (Appendix A), supporting RNA-Seq-based findings and interpretations. Targeted transcripts and respective primer sets used are shown in Appendix A.

### 3.5. Colocalization of DEGs in the Introgression Fragments

SNP genotyping revealed 16 N22-derived fragments in DTY-IL (Figure 10). The largest fragment was found on chromosome 1, encompassing *qDTY*1.1, followed by an introgression on chromosome 3, containing parts of *qDTY*3.2, which was also reported as N22-derived in an N22 by Swarna population [7]. Additional introgressions on chromosomes 4, 8, 9, and 10 did not overlap with major DTY QTL. Overlaying DEGs on the N22 introgressions identified 463 DEGs in the flag-leaf (Appendix A) and 433 DEGs in the panicle (Appendix A), of which 6 overlapped with the fine mapped region of *qDTY*1.1 [59], while 5 overlapped with the *qDTY*3.2 region.

Of the 6 DEGs in the *qDTY*1.1 region (Appendix A), LOC_Os01g67030 was upregulated in the panicle (Log_2_ fold change = 3.1), was annotated as an auxin-responsive protein of 418 amino acids (AA) in Nipponbare. While LOC_Os01g67030 was annotated in the *indica* reference MH63v2, it was missing in the N22v2 reference genome. FGENESH-based gene prediction in N22v2 revealed a putative homologue showing 91.4% identity with Nipponbare and 92.3% identity with MH63v2. Differences were found in the 5′UTR, resulting in the loss of coding sequence for the first 37 AA in the MH63v2 and N22v2 alleles, as well as a number of nonsynonymous (NS) SNPs (Appendix A). NS SNPs unique to N22 corresponded to four AA changes (P52L, C60F, S80T, Q137P) and an AA deletion (G253del) (Appendix A). While the alignment of the 2 kb upstream *cis*-regulatory region of LOC_Os01g67030 in Nipponbare and MH63v2 showed a 99.1% identity, N22v2 displayed a large deletion, including the 5′UTR. With an identity of less than 15% to either Nipponbare or MH63v2, the N22v2 *cis*-regulatory region of LOC_Os01g67030 was distinct with unique regulatory elements (Appendix A) that could explain the observed expression differences.

Of the 5 DEG within *qDTY*3.2, LOC_Os03g03510, downregulated in the panicle (Log_2_ fold change = −1.11617), was annotated as CAMK_KIN1/SNF1/Nim1_like.15-calcium/calmodulin-dependent protein kinase in Nipponbare, with corresponding annotations in MH63v2 and predictions in N22v2. LOC_Os03g03510 contains a sucrose non-fermenting 1-related kinase 3 (SnRK3) domain and a CBL-interacting serine/threonine-protein kinase 3 (CIPK3) domain. While the coding sequences were largely conserved in multiple sequence alignment between Nipponbare, MH63v2, and N22v2 alleles, the N22v2 allele featured an altered stop codon resulting in a 35 AA C-terminal extension (Appendix A).

## 4. Discussion

Source-sink relationships largely determine the grain yield of cereal crops, with developing grains being primary sinks, while the top two leaves, the flag-leaf, in particular, serves as the primary source [60,61]. Source sink regulation is orchestrated through intricate metabolic signaling [62], of which key players in sucrose sensing (e.g., trehalose-6-phosphate) and signal integration (e.g., SnRK1) are beginning to emerge [62]. Drought stress affects these relationships by reducing both source and sink strengths. In source organs, limitations in carbon fixation and primary metabolism lead to reduced resource allocation to developing sinks, causing yield reduction characterized by suboptimal grain filling [63]. In sink organs, drought reduces fertility, causing yield reductions through suboptimal seed setting [64].

While DTY-QTLs have demonstrated effects to improve rice grain yields under RDS, knowledge about underlying molecular mechanisms is limited. Functional studies of *qDTY12.1* suggested an intricate pattern of below-ground contributions [18], while physiological studies of *qDTY1.1* suggested above-ground implications [59]. Though confined to a single time point at the late booting stage (close to anthesis) after two weeks of RDS, our study suggested that DTY controlled mechanisms improve yield under drought by acting at both source and sink organs. At the flag-leaf, a coordinated response to sustain primary metabolism through cell wall loosening and maintained photosynthetic rates seems to allow for sufficient carbon and energy allocation to the developing panicle, which in turn enable reproductive structures to invest in protective mechanisms, including protein stabilization and turnover, ROS scavenging and production of protective secondary metabolites. While proximate effects in the panicle are suggested as improved male fertility and improved sink strength under RDS, the ultimate effects are improved seed setting and grain filling, and consequently, drought-tolerant yield (DTY) (Appendix A).

### 4.1. Source Effects—Flag-Leaf-Specific Differences between DTY-IL and Swarna

Collectively WGCNA and DE analyses suggested a complex interplay of a range of processes in the flag-leaf to contribute to the observed differences in RDS tolerance between DTY-IL and Swarna. These included specific protein turn-over, cell wall loosening, efficient ROS scavenging, and maintenance of photosynthesis (Figure 4, Figure 5 and Figure 6 and Appendix A).

A direct consequence of drought is impaired cell turgor [65], which is countered by the stiffening of cell walls to provide structural resistance [66,67]. Prolonged drought stress challenges plants to modify their cell walls, resulting in both cell wall tightening and loosening. Tightening occurs in tissues that are of relevance to structural integrity, while loosening occurs in tissues that need to be maintained in a growing and metabolically active mode [66].

Leaf rolling, a common indicator of drought stress in rice [68] was prominent in Swarna under drought but nearly absent in DTY-IL (Figure 1D). Leaf rolling likely relates to aberrant cell turgor and cell wall homeostasis and phenotypically reflects findings in the flag-leaf specific module M14.

Cell wall organization or biogenesis genes showed an increase in expression in FL-M14. A total of 12 cell wall-related genes were significantly upregulated in the DTY-IL and significantly downregulated in Swarna (Figure 6). These included two glycosyltransferase family 43 proteins, previously reported being involved in the synthesis of glucuronoxylan hemicellulose of secondary cell walls [69] and two expansin genes. Expansins facilitate loosening and extension of plant cell walls by disrupting non-covalent bonding between cellulose microfibrils and matrix glucans [70] and implications in response to dehydration are well documented [71,72,73,74,75] and rose [76].

Higher expression of cytoskeleton and cell cycle-related genes in DTY-IL (Figure 6) further supported the concept of maintained cell growth and stability in the tolerant flag-leaf tissue. Contrastingly, cytoskeletal genes (tubulin and formin) and a cell cycle gene (cyclin) were significantly downregulated in Swarna (Appendix A).

Several classes of enzymes control ROS production in the cell wall, most prominently plasma membrane NADPH oxidases [77] and class III peroxidases (CIII Prxs) [78]. CIII Prxs are secreted in the extracellular space, where they perform either cell wall stiffening through the peroxidative cycle [79] or cell wall loosening through the hydroxylic cycle [80,81]. In the present study, three CIII Prxs (LOC_Os03g13200, LOC_Os07g01370, and LOC_Os07g48020) were present in FL-M14 (Appendix A), with LOC_Os03g13200 and LOC_Os07g48020 significantly upregulated in DTY-IL and significantly downregulated in Swarna (Figure 6). High CIII Prxs expression in DTY-IL could support the generation of ^•^OH for cell-wall loosening through cleavage of cell wall polymers [67]. Interestingly, decreased expression of a calcium-dependent NADPH oxidase in Swarna and increased in activity of the DTY-IL in FL-M16 (Appendix A) was also observed. It is also known as respiratory burst oxidase, and is a well-studied enzymatic source of superoxide [82,83], which had previously been implicated in drought and high-temperature stability [83]. Hence, loosening of the cell wall and synthesis of structural constituents together is suggested to contribute to tolerance of water-deficit in the flag-leaf of DTY-IL.

More effective ROS scavenging, in general, seemed to be an important mechanism differentiating drought responses of Swarna and DTY-IL. Higher expression of peroxiredoxin in DTY-IL (Appendix A) suggests increased reduction capacity for H_2_O_2_, indicating enhanced detoxification in drought-stressed leaves.

A primary detrimental effect of water stress in source tissues is impaired photosynthesis [84]. Reduced abundance of photosynthesis-related proteins in response to RDS had previously been reported [85] and was indeed reflected in drought-stressed leaves of RDS-susceptible Swarna (FL-M16). Components of the light reaction (two photosystem genes, components of the core complex of photosystem II (PSII) involved the primary light-induced photochemical processes), the dark reaction (ribulose bisphosphatase and the fructose-1,6 bisphosphatase), and photorespiration (ribulose bisphosphate carboxylase large chain precursor) were found to be consistently downregulated in Swarna (Appendix A), suggesting functional impairments of general photosynthesis. Protection of photosynthesis from photoinhibition through photorespiration is a well-characterized drought response and furthermore prevents ROS accumulation in green tissues [86]. In addition, Swarna showed a lower expression of two ferredoxin-NADP genes, involved in thylakoid electron transport, suggesting reduced capacity in regulating the relative amounts of cyclic and non-cyclic electron for ATP and redox homeostasis [69]. Consequently, it is argued that the physiological environment in DTY-IL under RDS supports relatively higher rates of photosynthesis, which in turn, might sustain higher rates of energy and carbon production to support primary metabolism and source strength, ultimately leading to improved grain filling.

### 4.2. Sink Effects—Panicle Specific Differences between DTY-IL and Swarna

Collectively WGCNA and DE suggested a number of distinct mechanisms to contribute to differences in RDS tolerance between DTY-IL and Swarna in panicles. They included protein stabilization and turnover, ROS scavenging, biosynthesis of secondary metabolites for protection of reproductive organs, and hormonal signaling presumably to adapt reproductive developmental processes to drought. Under field conditions, they resulted in an approximate doubling of yield under drought for DTY-IL as compared to Swarna (Figure 1A), at no significant difference in plant height (Figure 1B).

Dehydration stress enhances the production of ROS and ROS-associated peroxidation causing damage to cellular structures [87]. Being essential for cellular signaling, ROS homeostasis depends on the balance between ROS production and scavenging [82]. Analysis of genome-scale metabolic pathways in the DTY-IL revealed the up-regulation of genes involved in the biosynthesis of antioxidant enzymes and metabolites (Figure 8 and Figure 9 and Appendix A).

Secondary metabolite production is crucial in stress-adaptive mechanisms [88]. Genes in pathways involved in secondary metabolite biosynthesis, lipid biosynthesis, redox homeostasis, amino acid metabolism, carbohydrate metabolism, and protein metabolism were upregulated at the maximum booting stage under RDS in DTY-IL and downregulated in Swarna for P-M10 and P-M15 (Figure 9). Several metabolic pathways found to be shared between the two modules include glutathione, terpenoid, and ascorbate metabolism.

De novo protein synthesis and turnover is fundamental for plants to cope with drought stress [85]. Translational efficiency is affected by ribosome composition, thus relative ribosomal protein abundance can modulate plant environmental responses [89]. Similarly, drought-responsive peptidases and heat shock proteins can alter the active proteome to cope with stress [85,90]. In P-M10 five ribosomal protein-related genes, six protein degradation-related genes (among them four peptidases), and two genes related to protein folding and repair displayed higher expression in DTY-IL (Figure 9; Appendix A). Congruently reduced expression of two peptidases and three genes related to protein processing, including a heat shock protein was observed for Swarna in P-M15 (Figure 9; Appendix A). Collectively this suggested that panicles of DTY-IL were more responsive and had the necessary energy to adapt its proteome to drought conditions than Swarna.

In P-M10, six genes involved in the ROS scavenging (two ascorbate peroxidases and four peroxidase precursors) had elevated expression profiles in the DTY-IL (Figure 9; Appendix A). Efficient reduction of H_2_O_2_ by peroxidases had previously been implicated with drought-tolerance in rice [19]. Specifically, plant ascorbate peroxidases (APXs) are crucial for ROS homeostasis [91] and free radical detoxification though the ascorbate-glutathione cycle [92], and their functional role in rice drought tolerance was demonstrated through transgenic approaches [91]. In P-M15 three ROS scavenging genes (1 glutathione *S*-transferase, 1 glutathione peroxidase, and 1 stromal ascorbate peroxidase) had a lower expression profile in Swarna (Figure 9; Appendix A). Glutathione peroxidase (GPX) catalyzes the reduction of H_2_O_2_ using thioredoxin (Trx), while glutathione *S*-transferases (GSTs) are key to the removal of xenobiotic compounds [85]. Ectopic expression of a GST in *Arabidopsis* [93] and a GPX in rice [94] resulted in enhanced drought tolerance, suggesting functional implications.

Interestingly, an ABC function gene with AP2 domain-containing protein (LOC_Os07g22770) controlling floral organ identity was downregulated in Swarna RDS under P-M15 (Appendix A), suggesting a link to aberrant Swarna floral development under drought [95]. TFs belonging to AP2 and MYB family are involved in panicle development as well as water-deficit stress response, implying that they may represent a crosstalk component between redevelopment and stress.

A negative regulator of plant drought tolerance in abscisic acid (ABA) signaling, protein phosphatase 2C (PP2C) [96], was upregulated in the DTY-IL in P-M10 (Appendix A). PP2C inhibits the activity of sucrose non fermenting 1 related kinase 1 (SnRK1) [97], a central integrator of metabolic signaling and regulator of starvation response. Thus, the higher expression of PP2C in DTY-IL might correlate with reduced SnRK1 activity, indicative of anabolism rather than catabolism and thus growth rather than the starvation mode in panicles of DTY-IL.

Brassinosteroids (BRs) are growth-promoting steroid hormones important for male fertility and pollen development [98]. BR catabolism is controlled by BAS1, a cytochrome P450 monooxygenase [99]. BRs bind to the extracellular domain of a cell-surface receptor kinase, BRASSINOSTEROID INSENSITIVE1 (BRI1) to activate kinase activity [100,101]. In P-M10, a BAS1-orthologue and two BRI1 genes were found to be upregulated in DTY-IL (Appendix A), suggesting a role for BR signaling in the maintenance of male fertility as part of the *qDTY*1.1-mediated RDS responses.

### 4.3. qDTY1.1-Specific Contributions to Drought Tolerance

Of the 16 N22-derived introgressions in DTY-IL, two overlapped with known *qDTY*s for which N22 was a known donor [21]. While the large introgressions on chromosome 1 contain the full *qDTY*1.1 region, a smaller introgression on chromosome 3 partially overlaps with *qDTY*3.2 (Figure 10). Working under the assumption that the genome-wide changes in transcriptomes between the drought-tolerant and drought susceptible genotypes in both flag-leaves and panicle, would at least partially trackback to either transcriptional or allelic differences of specific loci within the *qDTY* regions we took a closer look at the *qDTY*1.1 fine mapped region and *qDTY*3.2 overlap.

Based on the Nipponbare reference, the fine-mapped *qDTY*1.1 region encompasses 79 genes, of which six were differentially expressed between DTY-IL and Swarna under RDS in either tissue (Appendix A). An auxin-responsive protein (LOC_Os01g67030), specifically upregulated in DTY-IL panicle under RDS was considered as a likely causative candidate. Generally, auxin has been shown to negatively regulate drought adaptation in plants [63]. Notably, the DEEPER ROOTING 1 (DRO1) promoter was shown to contain auxin-responsive elements (AuxRes) and negatively regulated by auxin [102]. While two AuxRes were found in the LOC_Os01g67030 promoter of Nipponbare and MH63v2 (-513 bp and -1606 bp) only one was found at position -1573 bp in N22v2. The absence of the proximal AuxRe motif, in addition to the presence of novel, putative drought-responsive elements (Appendix A) could explain the observed differential regulation of LOC_Os01g67030 under RDS in DTY-IL.

LOC_Os01g67030.1 contains a cytochrome b561 (Cyt_b561) and a dopamine β-monooxygenase (DOMON) domain (Appendix A). Cyt_b561 proteins are involved in the regeneration of ascorbate through transmembrane electron transport [103,104] and have previously been implicated in drought tolerance through redox homeostasis [105]. The functionally uncharacterized β sheet-rich DOMON domain has been implicated in sugar and heme recognition [106] and predicted to be involved in protein-protein interactions, putatively functioning in metabolic signaling, in redox reactions, or both. Interestingly, LOC_Os01g67030 thus has the potential to link sugar signaling and ROS signaling, both of which have emerged as essential in the DTY-IL-specific drought response.

The 5 AA differences in the N22v2 prediction of LOC_Os01g67030 sequence fall under the two conserved domains (Appendix A). Notably they include two proline conversions and a glycine deletion, with potential structural implications, particularly in the context of transmembrane domains and β sheets [103,104]. This could affect the ability of the Cyt_b561 domain to mediate transmembrane transport and the DOMON domain to mediate protein-protein interactions or ligand binding.

Efficient ROS scavenging was identified as a key mechanism of RDS tolerance in both panicles and flag-leaves of DTY-IL. In the panicle, LOC_Os01g67030 could directly contribute to ROS homeostasis and with ROS being increasingly implicated in stress signaling including modulation of gene expression [107,108], LOC_Os01g67030 activity could be responsible for the some of the expression changes observed in P-M10 and P-M15. LOC_Os01g67030, however, was not expressed in flag-leaves. It is possible that the ultimate positive effects of the N22 allele of LOC_Os01g67030 on seed setting in DTY-IL panicles could increase sink strength in a way that it positively affects the source strength of flag-leaves, which could contribute to maintained photosynthetic rates. A similar sink on source effects has been demonstrated by manipulation of SnRK1 dependent metabolic signaling in maize under control and drought [109].

LOC_Os03g03510 was found downregulated in DTY-IL. Both CIPK_C domain [110] and SnRK3 domain [111,112] have been implicated in abiotic stress responses, including drought tolerance. In addition, SnRK3, like SnRK1 [62,109] has been demonstrated to function in metabolic signaling and source-sink relationships. In sinks, SnRK1 activity has detrimental effects on grain filling [109]. A 35 AA C-terminal extension in the N22 allele could have functional implications, which, in addition to its observed downregulation could reduce its efficiency in DTY-IL. The postulated effect would be altered downstream phosphorylation responses with potential transcriptional changes that reflect some of the differences seen between Swarna and DTY-IL Ultimately this could contribute to maintained sink strength of the panicle with putative effects on flag-leaf source metabolism.

In theory the postulated functions of both candidates could have synergistic effects that could explain a range of the observed DTY-IL specific drought responses. Gene validation studies expressing the N22 allele of LOC_Os01g67030, LOC_Os03g03510, or both, under control of their native promoters in drought susceptible *indica* background, respective knock-outs in DTY-IL, or both, are needed to confirm their postulated roles.

## 5. Conclusions

This study provided novel insight into global transcriptional responses in rice under moderate RDS in a DTY-dependent manner and highlighted associated physiological mechanisms that allow DTY-IL to better cope with RDS. In DTY-IL flag-leaves, structural and metabolic integrity associated with cell wall re-organization and active ROS metabolism prevent leaf rolling and allow for maintenance of cellular growth and homeostasis under RDS, which supports sustained rates of photosynthetic activity and consequently provisioning of energy and carbon to developing sinks. In the developing panicles close to anthesis, sustained energy and carbon allocation enables the minimizing of damage to reproductive structures due to RDS through protective mechanisms, including ROS homeostasis, post-transcriptional modifications, detoxification, and secondary metabolite production; ultimately this results in improved fertility and yield under moderate RDS (Appendix A). Assessment of DTY-specific allelic variation within the *qDTY1.1* and *qDTY3.2* regions prioritized two candidate genes in DTY-IL, a predicted auxin-responsive protein with a DOMON_DOH and a Cyt_*b561* domain, and a CIPK_C and SnRK3- domain-containing protein, which might positively affect source-sink regulation under drought.

## Figures and Tables

**Figure 1 genes-11-01124-f001:**
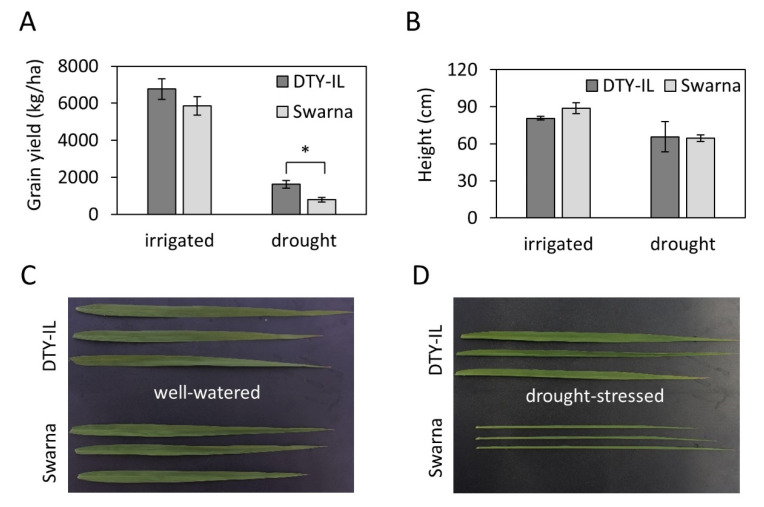
The effect of reproductive stage drought on yield, height and flag-leaf morphology. The average grain yield (**A**) and plant height (**B**) across the 2014 and 2015 dry season (DS) field trial of DTY-Il and Swarna under irrigated and drought condition (*N* = 2–9). Flag-leaf phenotypes of the DTY-IL and Swarna under well-watered conditions (**C**) and the leaf-rolling phenotype during drought-stress (**D**) under controlled greenhouse conditions. The asterisks (*) indicates a significant difference (Student’s *t* test, * *p* < 0.01).

**Figure 2 genes-11-01124-f002:**
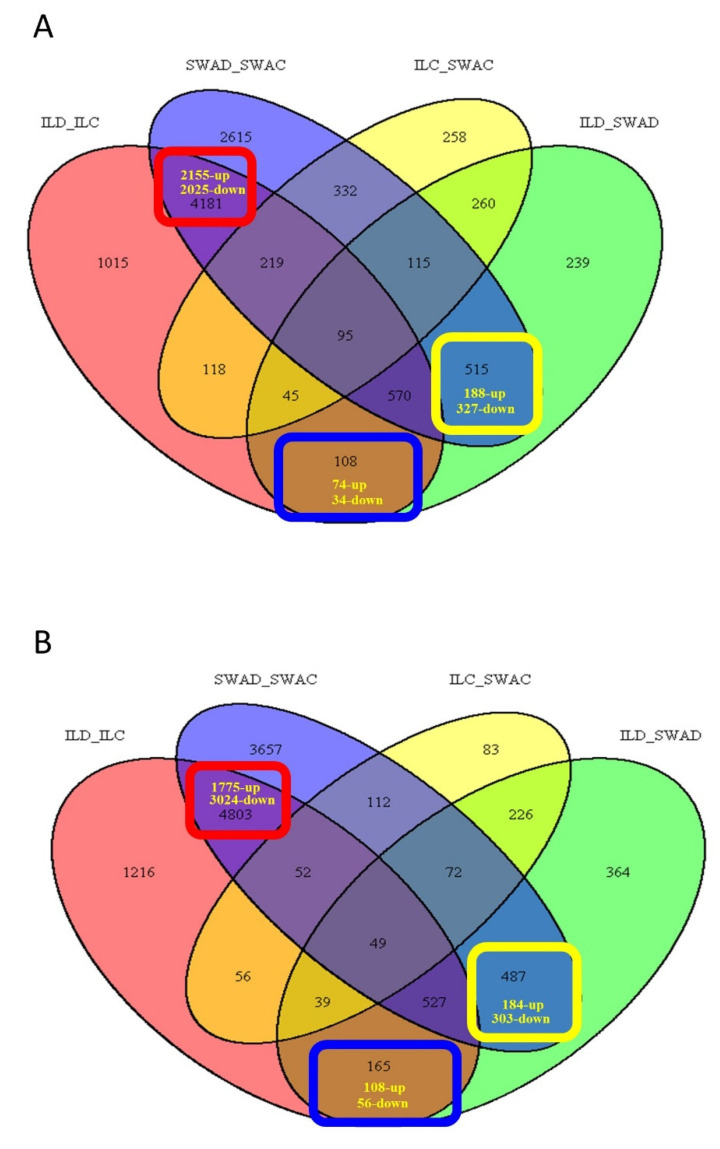
Gene differential expression and identification. Venn diagram of differentially expressed genes (DEGs) for flag-leaf (**A**) and panicle (**B**) tissues in Swarna and DTY-IL under reproductive drought stress (RDS) at a false discovery rate (FDR) adjusted *p*-value < 0.05 and −1 ≤ log_2_-ratio ≤ +1 (fold change ≥ 2 and ≤ 0.5). The three highlighted boxes for each tissue represent the common DEGs (red), unique to Swarna (yellow), and unique to DTY-IL (blue). SWAC = Swarna control, SWAD = Swarna under RDS, ILC = DTY-IL control, ILD = DTY-IL under RDS.

**Figure 3 genes-11-01124-f003:**
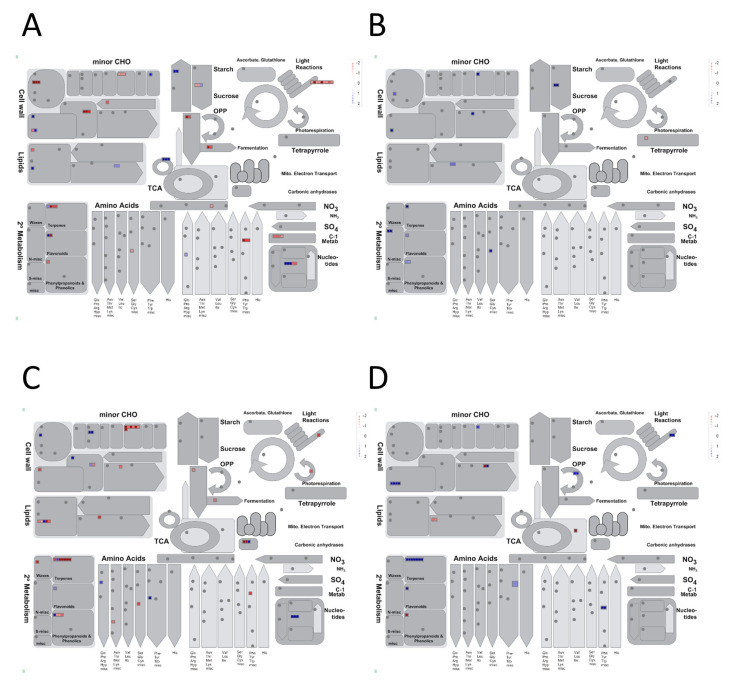
Mapman overview of the DEGs of interest in DTY-IL and Swarna under RDS. The metabolism overview in in the flag-leaf tissue for Swarna (**A**) and DTY-IL (**B**), as well as for the panicle tissue in Swarna (**C**) and DTY-IL (**D**). The DEGs were binned to the MapMan functional categories. The values are the log_2_ fold changes. Upregulated categories are represented in blue and downregulated categories in red.

**Figure 4 genes-11-01124-f004:**
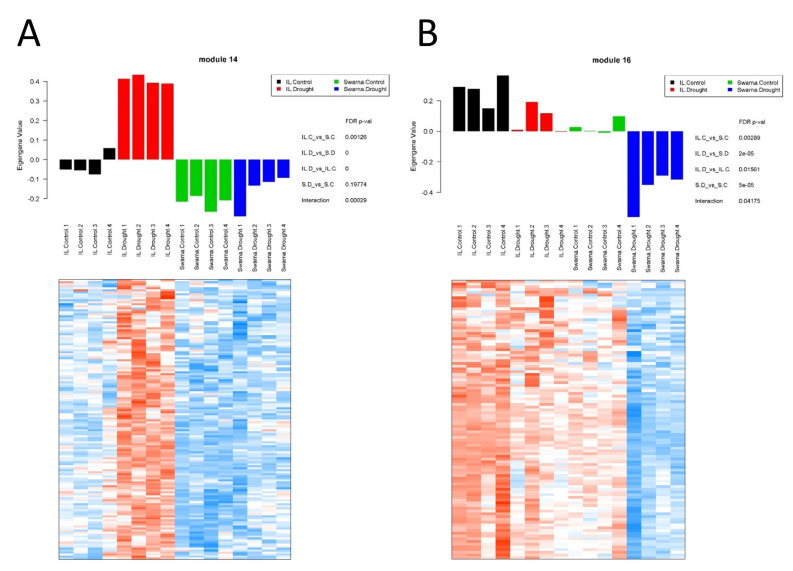
Gene co-expression network analysis in flag-leaf under RDS. Bar plots of the module eigengene as representatives of gene expression profiles across samples in FL-M14 (module 14) (**A**) and FL-M16 (module 16) (**B**). *X*-axis represents 16 different samples across four different groups, while *Y*-axis corresponds to the eigengene value. Heatmaps showing gene expression levels of genes in FL-M14 and FL-M16. Columns represent samples, while rows correspond to genes in the module. Red indicates positive and blue negative expression profile. S.C = Swarna control, S.D = Swarna under RDS, IL.C = DTY-IL control, IL.D = DTY-IL under RDS.

**Figure 5 genes-11-01124-f005:**
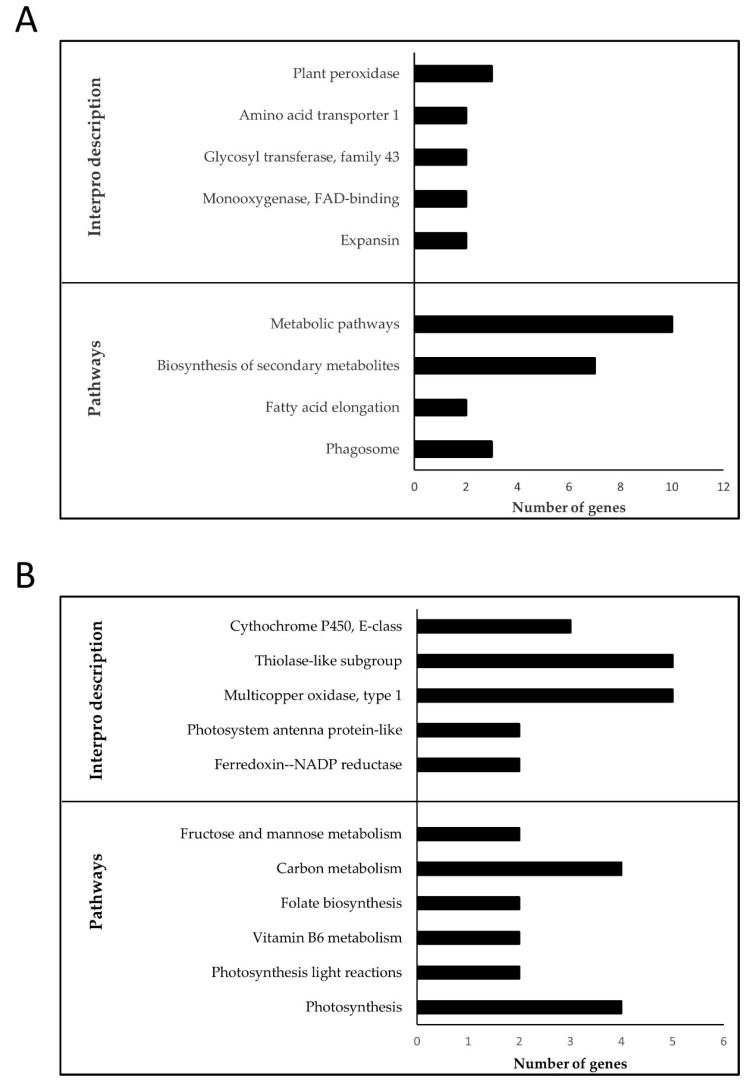
Enrichment analysis of the functional categories in FL-M14 and FL-M16. Over-represented Interpro domains and enriched pathways in FL-M14 (**A**) and FL-M16 (**B**) in flag-leaf under RDS. Top significant pathways and Interpro domains are shown in *Y*-axis with the number of represented genes on the *X*-axis.

**Figure 6 genes-11-01124-f006:**
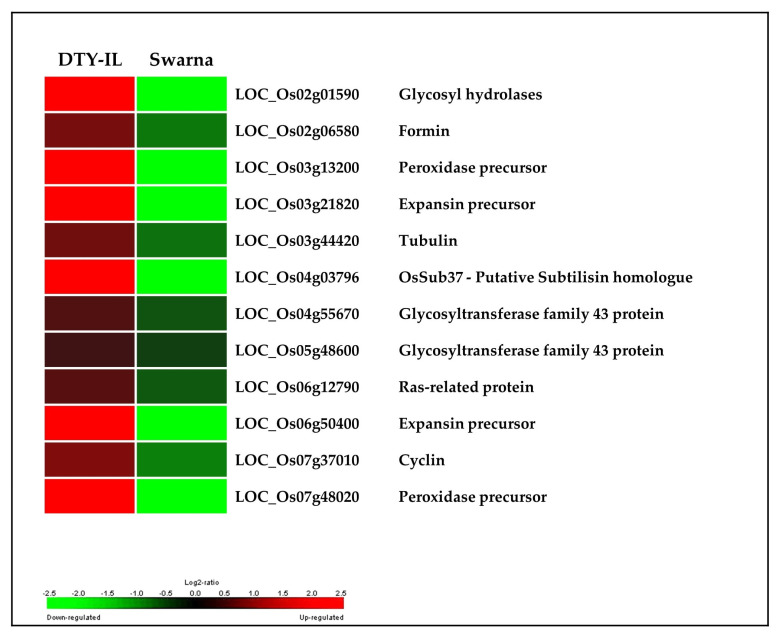
Cell wall organization and biogenesis during RDS in the flag-leaf tissue. Expression of cell wall organization or biogenesis related genes in drought-responsive modules between DTY-IL and Swarna are shown as log_2_ fold change heat-maps.

**Figure 7 genes-11-01124-f007:**
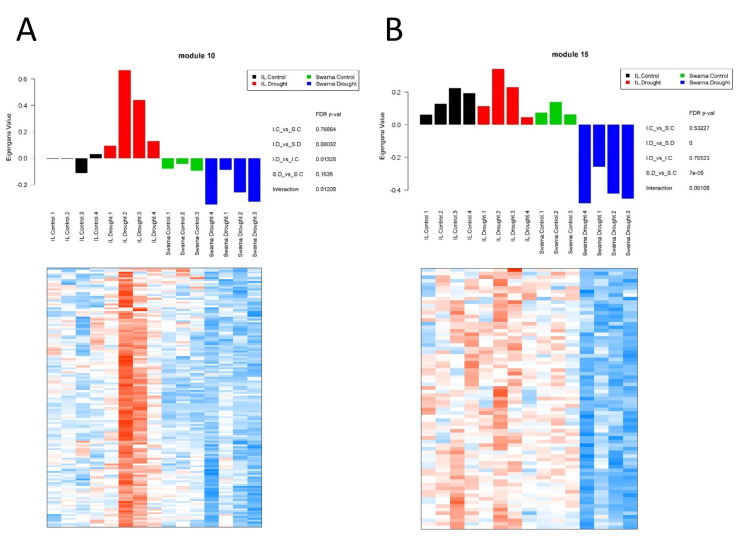
Gene co-expression network analysis in panicle under RDS. Bar plot of the module eigengene as representatives of the gene expression profiles across different samples in P-M10 (module 10) (**A**) and P-M15 (module 15) (**B**). *X*-axis represents 15 different samples across four different groups, while *Y*-axis corresponds to the eigengene value. Heatmaps showing gene expression levels of genes in P-M10 and P-M15. Columns represent samples, while rows correspond to genes in the module. Red indicates positive and blue negative expression profile. S.C = Swarna control, S.D = Swarna under RDS, IL.C = DTY-IL control, IL.D = DTY-IL under RDS.

**Figure 8 genes-11-01124-f008:**
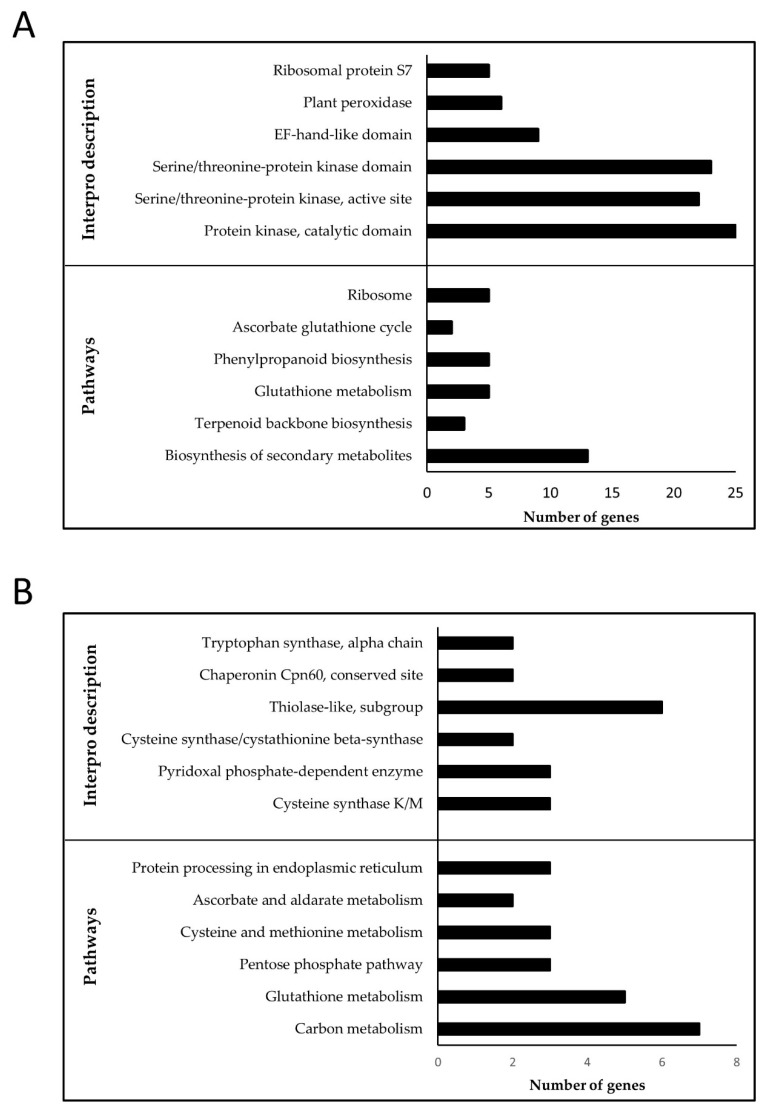
Enrichment analysis of the functional categories in P-M10 and P-M15. Over-represented Interpro domains and enriched pathways in P-M10 (**A**) and P-M15 (**B**) in panicle under RDS. Top significant pathways and Interpro domains are shown in *Y*-axis with the number of represented genes on the *X*-axis.

**Figure 9 genes-11-01124-f009:**
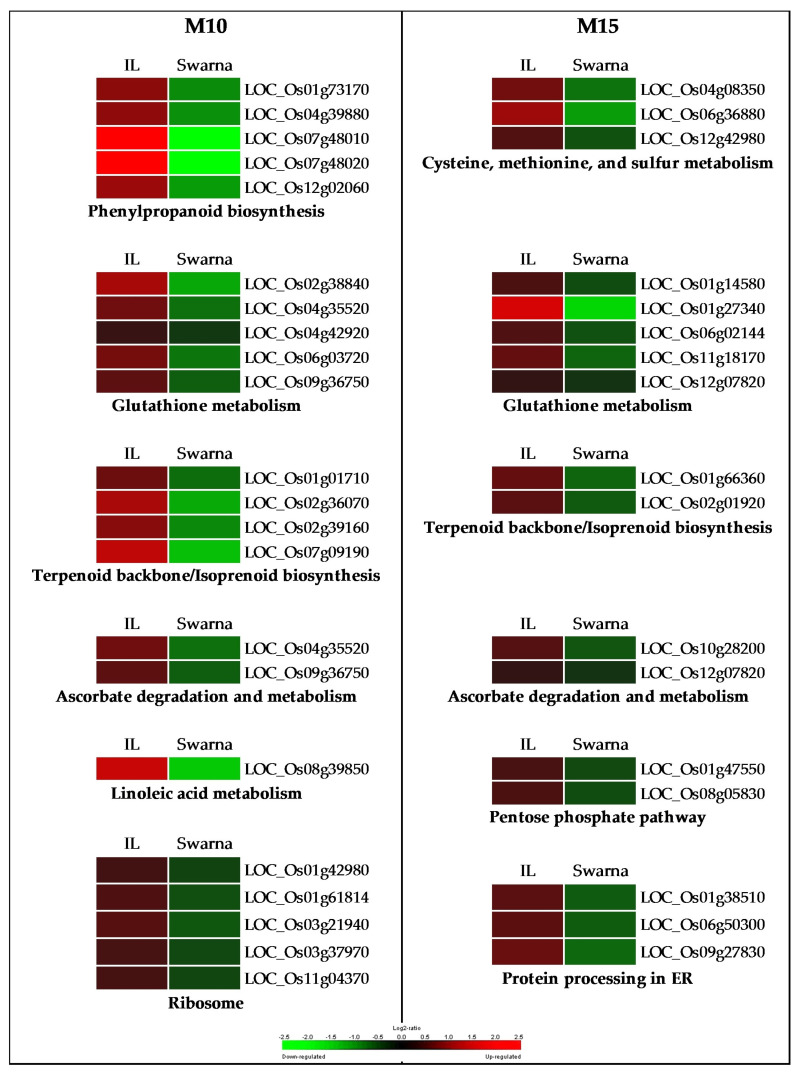
Regulation of metabolic pathways during RDS in panicle. The metabolic pathways enriched in the drought-responsive modules P-M10 (M10) and P-M15 (M15) between DTY-IL and Swarna under RDS are shown in heat-maps representing their expression profile. The scale represents a log_2_ fold change in expression. IL = DTY-IL.

**Figure 10 genes-11-01124-f010:**
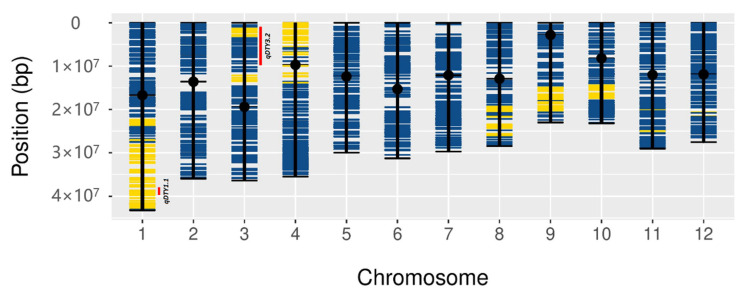
Physical map of DTY-IL. Genome-wide physical position distribution of 1648 SNPs from the 7K genotyping assay across all rice chromosomes. Swarna SNP’s alleles are represented in blue. N22 SNP alleles are represented in yellow. Names and ranges of N22-derived DTY QTLs (*qDTY*1.1 and *qDTY*3.2) are shown as red bars on the sides of the chromosome, more details are provided in Appendix A.

**Table 1 genes-11-01124-t001:** A summary of GO enrichment analysis of differentially expressed genes (DEGs) common to both genotypes, and unique to DTY-IL and Swarna for flag-leaf and panicle tissues under RDS.

DEGs of Interest	Expression	Genes		GO Terms		Significant GO Terms	
		Flag-Leaf	Panicle	Flag-Leaf	Panicle	Flag-Leaf	Panicle
Common responses of DTY-IL and Swarna DEGs to drought	Upregulated	2155	1775	409	324	88	110
	Downregulated	2025	3024	482	525	171	135
	Sub-total	4180	4799	891	849	259	245
Unique responses of Swarna DEGs to drought	Upregulated	188	184	92	89	0	0
	Downregulated	327	303	134	141	35	13
	Sub-total	515	487	226	230	35	13
Unique responses of DTY-IL DEGs to drought	Upregulated	74	108	31	58	4	36
	Downregulated	34	56	19	22	0	0
	Sub-total	108	164	50	80	4	36

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
