# Peer review of "Comparative Transcriptomics and Co-Expression Networks Reveal Tissue- and Genotype-Specific Responses of qDTYs to Reproductive-Stage Drought Stress in Rice (Oryza sativa L.)"

_genes, 2020, doi:10.3390/genes11101124_

Round 1

Reviewer 1 Report

Authors have used flag leaf and panicle tissues of a Drought-Tolerant Yield introgression line, and Swarna, to understand the expression and co-expression network at the reproductive stage in response to drought stress. The authors have applied an employed RNA-Seq, have done a great job in analyzing and interpreting the molecular results.  However, I did not understand the rationale of the study. Moreover, there are no clear hypothesis-driven objectives -this piece of information is missing. Another biggest downside of the article that I could see is, authors have not shown the phenotypic performance of selected in this experiment (there is no biological/trait data).

Comments to Authors 

  • It is certainly not clear how much of cumulative water transpired during the experimental periods or at least during the stress periods.
  • Authors should consider providing more information related to crop husbandry, treatments, and temperature pattern during the experiment in the greenhouse. The author should the field capacity maintained in the experiment with some of the visual (X-axis number of days and Y-axis as ‘field capacity’), this would help the reader to understand how the water deficit treatment was imposed and when.
  • L92-93: ….. “cultivar with its corresponding drought-tolerant introgression line, we were aiming to identify trait contributing alleles within know DTY QTL from the drought tolerance donor N22”. There is NO trait/biological data in the current version. The authors must consider providing actual trait data that was measured in the study. The reader would like to see the following in traits data between lines studies or also under treatments (flag leaf area, panicle length, spikelet number, spikelet fertility, and grain weight).
  • L108-109: Did the authors use the whole panicle for RNAseq? Or was it just a spikelet used? The authors must explain if they used the top or middle or bottom portion of the spikelet for their analysis. As authors would know, the position of the spikelet also significantly contributes to the differential expression.
  • L126: “An indexed transcriptome fasta file was built from the rice japonica genome (IRGSP1.0)”. It appears that authors aligned their RNA-seq data with the japonica genome and identified differentially expressed genes. Does the ability to detect

Reviewer 2 Report

This manuscript describes the integrating analysis of DEGs in response to drought between drought sensitive line and drought tolerant line. However, errors in writing style and grammar, and incomplete sentences hinder me to follow new findings in this study. Most of data are the level of preliminary data. First of all, I recommend intensive editing of this manuscript and most of figure data is needed to be improved. In addition, I prepared more comments to help improvement of this manuscript.

No1. Figure 1 well summarize results of all transcriptomes, but further analysis will not well be focused. I think that three groups highlighted from figure 1 is more effective than co-expression module analysis.

Next authors need present the enrichment analysis of UP and down regulated genes in three groups as shown in https://www.ncbi.nlm.nih.gov/pmc/articles/PMC6470995/. After then, authors can easily distinguish common or different GO biological processes among the DEG groups. Current GO and lacks the statistical supports. This analysis is needed to be replaced by enrichment analysis with p-value support and fold enrichment value of selected terms.

No2. For interpro domain pathway enrichment analysis, similar approaches are needed. This analysis can be replaced by MapMan analysis mapped by up DEGs and down DEGs of three groups, respectively.

No3. Key biological process are needed to be evaluated by additional experiments.

No4. The association of DEGs and SNP is very impressive and to clarify the candidate gens, authors need to validate the expression patterns under stress condition and validation of genetic variation in the candidate genes by sequencing analysis.

No5. Authors need to make the drought tolerance mechanism by summarizing all the data.

Round 2

Reviewer 2 Report

Authors mostly well addressed my comments but still need more improvement. module numbers after coexpression network analysis between flag leaf and panicle are confused. It will be better to distinguish them. For example, flag leaf modules, FM1, 2,....; panicle modules, PM1, 2, 3, ......

In GO enrichment data, some GO names in Y axis is incomplete and please make them complete. In addition, GO enrichment analysis usually does not allow one time identification for the selected GO term, interpro, and pathway for the statistic analysis. Please remove those GO in the graph and table data throughout text and supplemental data. Relating text should be modified. 

Figure S16 needs more explanation of labels in x and y axis. Figures S18-20 needs more explanation of MH63V2, Swarna proxy and N22v2

In line 797, it will be better to use in field condition than under field condition because under was used repeatedly in the same sentence.

Figure 10 needs more explanation on the colors in the chromosome map.

Although authors provide huge data, many of data have limit on the explanation of the data which makes it difficult to follow up the content. 
